# A Systematic Scoping Review of Indigenous People’s Experience of Healing and Recovery from Child Sexual Abuse

**DOI:** 10.3390/ijerph21030311

**Published:** 2024-03-07

**Authors:** Jordan Gibbs, Helen Milroy, Stella Mulder, Carlina Black, Catherine Lloyd-Johnsen, Stephanie Brown, Graham Gee

**Affiliations:** 1Intergenerational Health Group, Murdoch Children’s Research Institute, Melbourne 3052, Australia; stella.mulder@mcri.edu.au (S.M.); carlina.black@mcri.edu.au (C.B.); cat.lloydjohnsen@mcri.edu.au (C.L.-J.); stephanie.brown@mcri.edu.au (S.B.); graham.gee@mcri.edu.au (G.G.); 2Department of Psychological Sciences, Swinburne University of Technology, Hawthorn 3122, Australia; 3Division of Psychiatry, Faculty of Health and Medical Sciences, University of Western Australia, Perth 6009, Australia; helen.milroy@uwa.edu.au; 4Department of Paediatrics, University of Melbourne, Melbourne 3052, Australia; 5Centre for Community Child Health, Royal Children’s Hospital, Murdoch Children’s Research Institute, Melbourne 3052, Australia; 6School of Psychological Sciences, University of Melbourne, Melbourne 3000, Australia

**Keywords:** protocol, scoping review, Indigenous, healing, child sexual abuse

## Abstract

Child sexual abuse is a form of violence that occurs across nations and cultures. Collective efforts are being made to address this issue within many Indigenous communities. In Australia, Aboriginal and Torres Strait Islander communities have expressed the need for cultural models of healing child sexual abuse. A preliminary exploration of the relevant literature shows a lack of synthesis with regard to the current evidence base. This protocol outlines the methods and background for a scoping review that aims to explore and collate the broad scope of literature related to healing from child sexual abuse within an Indigenous context. The proposed review utilises a ‘population, concept, and context structure’ from the Joanna Briggs Institute to explore the broad scope of the literature within a scoping review framework. The target population is Indigenous survivors of child sexual abuse, including Indigenous populations from six distinct regions: Aboriginal and Torres Strait Islander peoples from Australia; Māori peoples from Aotearoa (New Zealand); First Nations, Inuit and Métis peoples from Canada; Native American peoples from North America; Native peoples from Alaska; and the Sámi peoples of the Sápmi region in Northern Europe. The concept within the review is healing from an Indigenous perspective, which includes a broad range of processes related to both recovery and personal growth. The contexts explored within this review are any context in which healing from child sexual abuse can occur. This may include processes related to disclosure and accessing services, specific interventions or programs for survivors of child sexual abuse, as well as broader non-specific healing programs and personal experiences of healing without intervention. The scoping review will use search strings with broad inclusion and exclusion criteria to capture the potential breadth of perspectives. The search will be conducted across several academic databases and will also include an extensive search for grey literature. This protocol establishes the proposed benefits of this scoping review.

## 1. Study Context

### 1.1. Background

Child sexual abuse is an urgent human rights and public health issue for all nations and cultures across the globe. For several years, a coalition of Aboriginal and non-Indigenous services in the state of Victoria, Australia have worked together carefully in a coordinated effort to address healing, recovery and prevention of child sexual abuse within Victorian Aboriginal communities. This collaboration between services builds upon decades of often unacknowledged work by Victorian Aboriginal Elders, leaders and community members who for years have been supporting Aboriginal and Torres Strait Islander survivors in this difficult and sensitive area. The coalition of services, funded by the Victorian government, has recently undertaken several important initiatives, including employment of a workforce specifically dedicated to supporting survivors and the design of training and education modules for health professionals working in this area of healing and recovery.

A small team of Aboriginal and non-Indigenous researchers from the Murdoch Children’s Research Institute will support the coalition of services by conducting a scoping review to identify the current evidence with respect to Indigenous survivors’ experiences of healing from child sexual abuse. Formative work leading to the development of this protocol has involved a series of meetings, consultations, and workshops with several of the respective services involved. Within that formative work, it was determined that it is important to explore the literature directly relevant to Aboriginal and Torres Strait Islander peoples, as well as literature exploring the worldviews and experiences of other Indigenous peoples around the world who have had similar historical and contemporary experiences, which can help inform the work of the services. The purpose of this scoping review is to identify the key factors associated with healing for Aboriginal and Torres Strait Islanders and other Indigenous survivors of child sexual abuse to inform future research, therapeutic practices, programs and policies. Four research questions underpin the scoping review:What are the barriers to and enablers of disclosure for Indigenous survivors of childhood sexual abuse?What are the barriers to and enablers of access and engagement with services for Indigenous survivors of childhood sexual abuse?How do Aboriginal and other Indigenous survivors of childhood sexual abuse experience healing and recovery?What are the cultural components of healing and recovery from childhood sexual abuse for Aboriginal and other Indigenous peoples?

### 1.2. Introduction

As part of a preliminary search of the international literature, the researchers identified more than 50 systematic literature reviews, scoping reviews and meta-analyses on child sexual abuse. Of those, only five specifically targeted Australian-only populations [1,2,3,4] with one review specific to responding to Indigenous sexual assault [5]. It is therefore clear that there is a need to review the literature exploring Indigenous understandings and experiences of healing from child sexual abuse.

Given the global scale and impacts of child sexual abuse, it is surprising that currently there is a lack of shared language and consistency across countries and nations with respect to how child sexual abuse is defined and conceptualised. For the purpose of this review, consistent with the World Health Organization [6], child sexual abuse will be defined as the involvement of a child in sexual activity that he, she, or they do not fully comprehend, is unable to give informed consent or is not developmentally prepared and cannot give consent, or that violates the laws or social taboos of society.

It is important to recognise that child sexual abuse affects all nations and communities worldwide, and to dispel myths that undermine the fact that Indigenous children are sexually abused by Indigenous and non-Indigenous people alike. Alongside this recognition, child sexual abuse and its ongoing impacts have been an area of concern for Aboriginal and Torres Strait Islander communities in Australia and many Indigenous communities worldwide [7,8] for decades. Several Australian state and territory reports involving consultations with Aboriginal communities that have a specific focus on child sexual abuse have been conducted (e.g., [9,10]). Internationally, numerous other independent and governmental reports at national, state, or territory level with a focus on Indigenous communities demonstrate similar focuses on sexual violence and child sexual abuse [7,8,11]. These reports recognise that no single factor can account for childhood sexual abuse that is experienced in some Indigenous communities, and they draw on various models (e.g., [12]) to identify multiple risk factors that operate at the individual, family, community and national levels. International research demonstrates that across cultures and societies, child abuse is disproportionately reported among families and communities that experience social inequalities and/or political inequality and systemic discrimination, including factors such as poverty and unemployment, poor health, low education attainment, high incarceration rates, racism, and poor access to services (e.g., [13,14,15]).

All of these factors are relevant for Aboriginal and Torres Strait Islander communities and Indigenous communities worldwide, as are other factors linked to systemic violence associated with colonisation, such as the dispossession of land and resources, assimilationist forced child removal and institutionalisation policies, and the subsequent breakdown of kinship systems, traditional culture and lore. Due to these past and ongoing unresolved social justice issues, poor relationships with the police and welfare agencies are common within Aboriginal communities [16]. The mistrust and reluctance of some communities to engage with government agencies mean that the rates of sexual assaults may be higher than some reports suggest (for detailed reviews of violence and trauma in Aboriginal communities, see [17,18,19]). Although the data are limited, there are some data on the extent and impact of child sexual abuse in Aboriginal and Torres Strait Islander communities in Australia. According to data collected by the Australian Bureau of Statistics, in 2020, the rate of sexual assault reported to police among Indigenous children aged 0–14 years in New South Wales, Queensland, South Australia and the Northern Territory was more than twice the rate of reported sexual assault among non-Indigenous children in these jurisdictions (345 reported sexual assaults per 100,000 Indigenous children compared to 170 reported sexual assaults per 100,000 non-Indigenous children) [20].

In Australia, efforts to understand and address the impacts of child sexual abuse within Aboriginal contexts have been mixed. For example, the Australian government’s development and enactment of the Northern Territory Emergency Response following the release of the Ampe Akelyernemane Meke Mekarle *Little Children are Sacred* report [10] has been highly contested both nationally and abroad [21]. Whilst there was some increased provision of essential services during the intervention, Aboriginal, Torres Strait Islander, and non-Indigenous critics have argued that the Northern Territory Emergency Response violated basic human rights for Aboriginal people in the Northern Territory. For example, the 2010 United Nations Special Rapporteur report on the situation of human rights and fundamental freedoms of Indigenous people concluded that the Northern Territory Emergency Response “was racially discriminatory and incompatible with Australia’s international human rights obligations, and that the Racial Discrimination Act 1975 needed to be reinstated” [22].

Other significant large-scale initiatives in Australia, such as the Royal Commission into Institutional Responses to Child Sexual Abuse [23], which involved many Aboriginal and Torres Strait Islander survivors, have received positive responses from Aboriginal and Torres Strait Islander organisations. The Victorian Aboriginal Child Care Agency (VACCA) designed and delivered a cultural healing program to support Aboriginal survivors who came forward to share their story with the Royal Commission. This was based on survivors’ requests for healing that supported their long-term health and well-being needs and the understanding that clients “either did not access mainstream counselling or where it was accessed, it was insufficient for their healing” [24] (p. 1060).

The values and principles underpinning this cultural healing program were that any program designed to support Aboriginal people’s well-being needed to consider connections to identity, culture, family, community, ancestors and spirituality. The program included activities such as separate men’s and women’s business (i.e., ceremonies); activities to strengthen identity and connection to community; activities related to self-care; healing and well-being activities; sharing knowledge about past policies and histories of removal; discussing the impacts of removal and historical losses; storytelling and ‘yarning’; and sharing of meals [24] (p. 1067).

Black, Frederico and Bamblett (2019) [24] conducted a program evaluation of the cultural healing program and identified several core elements, which included: establishing psychological and cultural safety; the impact of culture in the healing journey (including connection to family, land, spirituality, and cultural activities); engaging the broader community (a whole-community approach including close kinship networks and localised community services and support); survivor empowerment; and understanding past, present and future healing. The authors argue that successful healing programs for Aboriginal child sexual abuse survivors require careful, safe community development processes that can be grounded in evidence-based practices.

The core elements identified by Black, Frederico and Bamblett (2019) [24] are consistent with a cultural framework for addressing child sexual abuse in Aboriginal and Torres Strait Islander communities developed by Milroy, Lawrie and Testro (2018) [25] on behalf of the National Aboriginal and Torres Strait Islander Healing Foundation that included similar key elements, including community-led responses; a holistic approach; connecting to cultural values and systems; healing; justice and reparation; and knowledge creation and sharing. The National Aboriginal and Torres Strait Islander Healing Foundation has defined healing as:

*Healing is not an outcome or a cure but a process; a process that is unique to each individual. It enables individuals, families and communities to gain control over the direction of their lives and reach their full potential. Healing continues throughout a person’s lifetime and across generations. It can take many forms and is underpinned by a strong cultural and spiritual base* [26].(p. 1)

We note some potential distinctions between resilience, recovery and healing outcomes. For example, prospective studies investigating outcomes for people 12 months after experiencing potentially traumatic events have found resilience trajectories to be marked by minimal impact, whereas recovery trajectories included elevated posttraumatic stress-related symptoms and functional impairment, followed by a gradual return to previous levels of functioning [27]. Within an Indigenous context, Professor Helen Milroy (2009), an Aboriginal psychiatrist, has made a distinction between recovery and healing:

*Healing is not just about recovering what has been lost or repairing what has been broken…It can be experienced in many forms…Mostly, however, it is about renewal. Leaving behind those things that have wounded us and caused us pain… with hope for the future, with renewed energy, strength and enthusiasm for life* [28].(p. 522)

This understanding of healing includes experiences of recovery (i.e., a return to baseline functioning and reduction in trauma symptoms of distress), but also speaks to other processes of renewal and growth. For this scoping review, we define healing as the experiences of survivors impacted by child sexual abuse that include both processes of recovery and personal growth.

There is considerable cultural diversity within different Indigenous groups and communities. In addition, individual survivors of child sexual abuse may have unique healing needs. For these reasons, there is likely to be variation in the core elements of programs and frameworks that focus on healing from child sexual abuse. As such, there is a need for a systematic examination of the current evidence base for Indigenous healing programs that aim to support survivors of child sexual abuse. Whilst common themes and elements are likely to be identified in the literature, some may have stronger evidence for successful implementation than others. Contextual factors, such as survivor characteristics (e.g., gender, age), intervention type, geographical location, and community settings, may all influence and have implications for the ways in which healing from child sexual abuse is addressed.

Currently, there are no English-language reviews on this topic to the best of the research team’s knowledge. The literature on healing from child sexual abuse in Aboriginal and Torres Strait Islander communities is also relatively limited. As such, there is a need for an international Indigenous-focused scoping review of the current literature to help identify gaps in current knowledge, support future research, and better inform the development of effective therapeutic practices, programs and policies in relation to healing from child sexual abuse. This protocol outlines the methods of a proposed scoping review that will privilege Aboriginal and Torres Strait Islander voices and experiences that address healing from child sexual abuse.

## 2. Method

This protocol, which details the methods for the proposed scoping review, was informed by and structured according to the Joanna Briggs Institute (JBI) Manual for Evidence Synthesis [29]. The items of this protocol are presented in accordance with relevant sections of that guide.

### 2.1. Research Design

This protocol details the design and procedures of a scoping review that will be conducted to explore some of the key factors associated with healing for Indigenous peoples who are survivors of child sexual abuse, with a particular focus, where possible and available within the literature, on Aboriginal and Torres Strait Islander peoples. All Indigenous groups, where possible, will be reviewed with consideration of their unique social, cultural, political, and geographical contexts, and discussed in accordance with this consideration. This review will identify the relevant literature that explores healing for Indigenous survivors of child sexual abuse across several contexts. This may include but is not limited to: healing through individual or group therapeutic programs or interventions; community or institutional healing programs/interventions; barriers to and enablers of healing; barriers to and enablers of disclosure; and survivor experiences of healing from child sexual abuse with or without targeted intervention. The review utilises the ‘population/participants, concept, context’ criteria to categorise articles and determine eligibility criteria as outlined in the Joanna Briggs Institute scoping review guidelines [29]. The ‘population/participants, concept, context’ criteria will allow for wide inclusivity to ensure the collation of relevant materials.

### 2.2. Eligible Types of Sources

In accordance with the Joanna Briggs Institute guidelines, this review will consider multiple sources, including academic and grey literature that meets the inclusion and quality criteria. Sources incorporating Aboriginal, Torres Strait Islander, and other Indigenous voices, experiences and knowledge will be prioritised. The review will exclude study designs that do not present unique findings or conclusions, including protocols and reviews that are not full versions of a study, abstracts, and conference presentations. Due to the historic exclusion of Indigenous voices and perspectives, the review will include a large variety of sources and aim to privilege Indigenous voices. Full inclusion and exclusion criteria can be seen in the Appendix A.

### 2.3. Eligible Study Participants/Populations

Eligible populations for this scoping review are defined as any persons or populations who are Indigenous and are survivors of child sexual abuse. This review will include Indigenous populations from six distinct regions: Aboriginal and Torres Strait Islander peoples from Australia; Māori peoples from Aotearoa (New Zealand); First Nations peoples, Inuit and Métis peoples from Canada; Native American peoples from North America; Native peoples from Alaska; and the Sámi peoples of the Sápmi region in Northern Europe (inclusive of Sweden, Norway, Finland and Russia). These participant/population groups may include any age, gender identification, sexual identification and other demographic variation within the inclusion criteria.

### 2.4. Eligible Study Concept

To be included in the study, a relevant document must have a specific focus on Indigenous peoples’ experiences of healing from child sexual abuse, including but not limited to associated barriers and enablers, factors and processes that inform healing, and the intersection of these factors or processes with broader social and emotional well-being, cultural conceptualisations of healing, and/or related interventions. Documents will be excluded if they do not inform the scoping review’s understanding of healing from child sexual abuse for Indigenous peoples.

### 2.5. Eligible Study Context

This review will broadly capture contexts in which Indigenous peoples experience healing from child sexual abuse. Healing can occur at any age and over any timeframe and may refer to any stage of the recovery or personal growth experience outlined in the above definition and be in scope for inclusion. This includes, but is not limited to: targeted/focused interventions (individual and group); broader well-being and healing interventions that specify healing from child sexual abuse as an outcome; community programs and evaluations; institutional programs and evaluations; and case studies and reports.

### 2.6. Search Procedures for the Identification of Studies

Academic literature will be searched using comprehensive publication searches undertaken using electronic databases Scopus, Medline and CINAHL via EBSCO, and PsycInfo and Embase via Ovid. The proposed search string (examples using appropriate Boolean and proximity operators for Scopus) is:(Indigenous OR Aborigin* OR {Torres Strait Islander} OR Māori OR Maori OR {First W/(nation* OR people* OR Australia* OR Canad* OR Alaska* OR America*)} OR {Native W/(Canad* OR Alaska* OR America*)} OR {America Indian} OR {Indian Country} OR Inuit OR Metis OR Métis OR Sámi OR Sami) AND(Child* PRE/(Sex* OR “Sex* Abuse” OR “Sex* Assault” OR “Sex* Offen*” OR Molest*)) OR (Survivor PRE/2 (Child* PRE/(Sex* OR “Sex* Abuse” OR “Sex* Assault” OR “Sex* Offen*” OR Molest*))) OR (Healing PRE/2 (Child* PRE/(Sex* OR “Sex* Abuse” OR “Sex* Assault” OR “Sex* Offen*” OR Molest*))) OR (Resilience PRE/2 (Child* PRE/(Sex* OR “Sex* Abuse” OR “Sex* Assault” OR “Sex* Offen*” OR Molest*))) OR (Coping PRE/2 (Child* PRE/(Sex* OR “Sex* Abuse” OR “Sex* Assault” OR “Sex* Offen*” OR Molest*))) OR (Recover* PRE/2 (Child* PRE/(Sex* OR “Sex* Abuse” OR “Sex* Assault” OR “Sex* Offen*” OR Molest*))) OR (Program* PRE/2 (Child* PRE/(Sex* OR “Sex* Abuse” OR “Sex* Assault” OR “Sex* Offen*” OR Molest*))) OR (Interven* PRE/2 (Child* PRE/(Sex* OR “Sex* Abuse” OR “Sex* Assault” OR “Sex* Offen*” OR Molest*))) OR (Treat* PRE/2 (Child* PRE/(Sex* OR “Sex* Abuse” OR “Sex* Assault” OR “Sex* Offen*” OR Molest*))) AND(Barrier PRE/2 (disclos* OR engag* OR access*) OR (Facilitat* PRE/2 (disclos* OR engag* OR access*) OR (Enable* PRE/2 (disclos* OR engag* OR access*).

Search strings with appropriate Boolean and proximity operators for each database can be found in Appendix B.

The search string was developed iteratively by utilizing terms identified within a preliminary search of the international literature. The researchers identified more than 50 systematic literature reviews, scoping reviews and meta-analyses on child sexual abuse from which recurring terms were drawn, alongside terms identified within broader Indigenous literature. Subsequently, each term was tested within a pilot search string and removing words individually and comparing the returned number of results. If the removal of a given term did not impact the number of records retrieved, it was determined redundant and excluded from the final search string. Variations of commonly accepted terms for Indigenous peoples from the selected geographic regions will be included to broaden the scope of the review. Truncation of terms will be used to allow for the inclusivity of relevant variations across population types, contexts and concepts. This search strategy will provide a broad platform for the identification of relevant articles and reduces the risk of relevant articles being missed by the search parameters.

Search filters will be applied to narrow the search parameters according to the exclusion criteria. The filters that will be used across all databases are relevant variations of the English language and geographical restrictions to the six noted regions. In addition to the multiple databases listed above, grey literature searches will be conducted in Google and Indigenous resource repositories and sources (e.g., The Australian Indigenous HealthInfoNet and Healing Foundation). All grey literature will undergo the same assessment procedure as academic literature. Grey literature searches will be international in scope and include all six geographical regions. Grey literature searches conducted in large databases or search engines, and will include, where possible, a ‘must include the phrase Child Sexual Abuse’ as a limiter. Grey literature searches conducted in large databases or search engines with results exceeding 20 pages will be sorted according to relevance and screened for no more than 20 pages. Furthermore, all sources included within the full-text screening will have their reference lists screened via a backward citation process for inclusion and also forward citation searching undertake to ensure this review can establish a comprehensive base of literature. Further relevant literature known to the research team will be screened and reviewed using the same process. Searches will be conducted a minimum of three times: an initial search, a search prior to data analysis and a final search before writing results. Searches will also be conducted every twelve months from the most recent search until any results are submitted for publication.

### 2.7. Screening and Selection of Eligible Studies

In accordance with the Joanna Briggs Institute guidelines, the screening will be conducted over three stages: title screening, abstract screening and full-text screening. All relevant articles will be imported into EndNote 20 (Clarivate, Philadelphia, USA) for collation and review. Before screening, EndNote’s deduplication function will be used to remove duplicate results. After deduplication, three researchers (JG, GG, and CLJ) will assess all titles against the inclusion and exclusion criteria. As detailed in the inclusion and exclusion criteria provided in the Appendix A, during the title screening step, titles are only to be screened with clear identification of a partial focus on three of the ten identified criteria, which allows for broader ambiguity to ensure documents are not excluded based on little information. This step is in place to exclude articles that clearly do not have any relevance to the review purpose. Three researchers (JG, GG, and CLJ) will evaluate all remaining abstracts to determine their relevance to the current study based on all inclusion and exclusion criteria. Once all excluded articles are discarded, full texts will be retrieved for the remaining included literature.

Five researchers (JG, GG, CLJ, CB and SM) will be assigned a selection of the full-text articles to read and evaluate independently while taking notes relating to inclusion and exclusion during this process. To determine consistency, all five researchers will read and evaluate the same three included full-text articles, and evaluations will be compared and discussed. Discrepancies in inclusion and exclusion decisions within those three articles will be discussed as a research team, and adjustments in the review process will be made as needed to best maintain inter-reviewer consistency.

Any discrepancies identified across all stages will be discussed with two other team members (HM and SB). The resolution of discrepancies will be achieved by consensus.

### 2.8. Data Collection and Extraction from Eligible Studies

Five review authors (JG, GG, CB, CLJ and SM) will independently extract data for a group of allocated literature using a pilot standardised data collection/extraction process in the web-based application Research Electronic Data Capture (REDCap) (Vanderbilt University, Nashville, TN, USA), which is designed to capture and securely store electronic research data. Data will be extracted under the following headings: Study Information, Data Extraction, Study Design/Methods, Interventions (if relevant), Outcomes and Quality Appraisal. Any information that is not included will be noted. In accordance with Joanna Briggs Institute guidelines, the provided sheet may be modified, as the extraction process is inductive and will vary depending on the type of literature identified within the search and screening process.

*Study designs* are considered to be any information that informs the type of data (qualitative, quantitative, mixed method), the closest reasonable description of the design (e.g., cross-sectional, case study, cohort study, randomised controlled trial, etc.) and any other information that informs the overarching design of the study, and would not best fit in another category (e.g., study methods).

*Study identifiers* are considered to be any information relevant to source the article, including: the names and affiliations of contributing authors; study title; publication type; date of publication; source information (for example, journal name, volume number, issue number and relevant page numbers); place of publication; and digital object identifier. Information regarding study design will include: the study type; relevant methods; data type and saturation; location/s of analysis; and relevant ‘population/participants, concept, context’ criteria.

*Study aims/hypothesis* are considered to be what the study identifies as its proposed logic, argument, thought or questions that broadly inform the scope and direction of the study. For sources that do not include any statements of direct aims or hypotheses, the authors may consider statements that broadly define the purpose of the study.

*Data extraction regarding participants* will include the target populations (e.g., Indigenous survivors of child sexual abuse), sample size, source of samples (e.g., clinical service, community program), recruitment methods and setting.

*Data extraction regarding intervention* will include the types of processes that informed the healing of participants, for example, any barriers or enablers that impacted disclosure, the type of healing process, the service or program being assessed, and when the intervention occurred.

*Data extraction regarding outcomes/results* and key findings will include any information that—based on the methods and procedures of the study—informs the healing of Indigenous survivors of child sexual abuse and relevant factors related to the provision of services. These sections will include considerable variation based on the type of study design and methods used across the included sources. As such, more detailed methods of extraction for quantitative and qualitative data are listed below. Results will focus on information that is presented as a direct result of study procedures.

*Other relevant information* will include any extracted information the authors believe to be relevant to the scoping review that does not fit under the above headings. Potential information in this section includes identified limitations, biases, or conflicts that are immediately identifiable within the document. This is not considered a formal evaluation of quality or bias.

### 2.9. Quantitative Data Collection and Extraction from Eligible Studies

Quantitative data from relevant studies will include in the relevant data extraction sections any information that supports an understanding of the healing of Indigenous survivors of child sexual abuse and relevant factors related to the provision of services. Examples may include study design (e.g., consultation statistics), participant characteristics (e.g., demographics—age, gender, etc.), methods (e.g., measure validation statistics) and outcomes (e.g., effect sizes).

### 2.10. Qualitative Data Collection and Extraction from Eligible Studies

Data from relevant qualitative studies will include in the relevant data extraction sections any information related specifically to the research question, including any description, anecdotal account, theoretical evidence, or analysis that supports an understanding of the healing of Indigenous survivors of child sexual abuse and relevant factors related to the provision of services.

Qualitative data will be synthesised by five researchers (JG, GG, CB, CLJ and SM), who will independently read and reread allocated eligible articles to inclusively and inductively assess key points and ideas. Notes and summaries of articles will be organised within the extraction database in REDCap. Qualitative data will be recorded according to the individual researcher’s inductive, subjective assessment of the article consistent with the synthesis process. All extracted data will be recorded and compared for inter-rater reliability during the comparison process as part of the synthesis of all extracted data.

### 2.11. Risk of Bias and Quality Assessment for Eligible Studies

As this review includes a broad scope of study designs and sources outside academia, assessment of methodological quality or risk of bias will not be conducted.

### 2.12. Quantitative Data Synthesis and Analysis Strategy

Quantitative data will be collated, organised descriptively, and presented in tables and figures. Studies deemed appropriately similar in methodologies and purpose will be compared based on effect sizes, demographics, study characteristics and key findings. No further quantitative analysis, such as meta-analysis, will be conducted.

### 2.13. Qualitative Data Synthesis and Analysis Strategy

#### Qualitive Data Synthesis

It is not feasible or possible to integrate the full scope of all the knowledge, experiences, histories, and activities of the First Nations groups identified in the literature search. As such, the research team acknowledge that this is a limitation. Qualitative thematic analysis of the literature identified across First Nations groups is beyond the scope of this review. Rather, it aims to descriptively locate, extract and map key findings as presented in the literature across the four areas of interest. Wherever possible, the descriptive synthesis will aim to maintain the accurate representation of the article without analysis or undue interpretation. This approach was specifically chosen to reduce the possibility of mischaracterisation or misinterpretation of the ways of knowing, being, and doing of Indigenous communities to which the research team does not belong. Where possible, the research team led by several Australian Aboriginal researchers will seek to contact any authors—notably First Nations authors and/or Elders associated with identified papers—when there is an identified need to seek clarity, and better understand and integrate findings, particularly with regard to highlighting any important cultural information and content.

Following initial extraction using the summary sheet, the five researchers (JG, GG, CB, CLJ and SM) will develop notes and summaries using an inferential process of data extraction [30]. This process will require the reviewers to immerse themselves in the literature by reading and rereading articles and continuously updating their summaries. A synthesis of the summaries of relevant articles will enable a collated understanding of the body of literature in terms of relevance in relation to the primary research questions.

### 2.14. Strategy for Presentation of Expected Results

The final search strategy and additional searches will be provided either in text, tables, or appendices in the final scoping review report. A flow chart following the PRISMA guidelines [31] will be used to illustrate the citation collection, screening, and exclusion processes for all searches as well as summarised reasons for exclusion. It is expected that the presented search and screening strategy will likely result in a complex mixture of peer-reviewed and grey literature. In addition, the identified literature is likely to span multiple First Nations peoples’ perspective worldwide with respect to the child sexual abuse discourse, and this means the research team will necessarily need to grapple with the challenges of sensitively capturing or documenting elements of cultural diversity. At the same time, it is also expected that the literature will include types of impacts and experiences associated with histories of colonization, dispossession, forced assimilation and other colonial harms that are common enough between First Nations communities worldwide to provide findings of value with regard to First Nation survivors’ experience of healing child sexual abuse. The final collection of included articles will be displayed in table format, including relevant study identifiers such as author names, year of publication, article title and study design. Demographic information will be displayed in a table including the number of participants within each study and information relevant to the study, such as age and gender of study participants. Study characteristics will be displayed in tables sorted according to study design type in the final article. The tables will include summarised versions of information on the data extraction database in REDcap and will include the same headings.

The review will incorporate narrative and descriptive commonalities between articles developed from the synthesis of the literature included. A table of a summary of key findings will be provided, including summarised key findings for each reviewed study.

## 3. Ethical Considerations

This review was conceptualised in response to requests from Victorian Aboriginal community organisations working with survivors of child sexual abuse. These organisations recognised the need for a stronger evidence base to inform their work. This research is being conducted by a research team consisting of three Aboriginal researchers and three non-Indigenous researchers and will actively follow the guidance of cultural protocols in support of Aboriginal data sovereignty, research governance and community safety [32]. The community consultation processes and development of this protocol have drawn on Aboriginal and Torres Strait Islander core values, rights regarding participating in research, and eight steps of the research journey, as outlined by the Australian National Health and Medical Research Council [33]. The research team prioritised Victorian Aboriginal people’s worldviews, right to self-determination, and right to have input into the research agenda by conducting one-to-one and group consultations with the respective organisations, in order to identify priority research areas and needs from an Aboriginal health professional and community perspective. Hence, early project steps included building relationships, developing the research idea, and developing the project and seeking agreement from relevant members of the Victorian Aboriginal community [32]. These cultural and ethical research protocols will be integral to completing the scoping review through maintaining relationships and meeting with the organisations at key stages of the review. This will provide regular opportunities to share understandings, receive feedback about the findings, and explore how the findings can best be utilised, translated and aligned with organizational aspirations to design Aboriginal sexual assault services. Additionally, as this work is being undertaken by a team at the Murdoch Children’s Research Institute, this project will be discussed regularly by the Aboriginal and Torres Strait Islander Health Research Program Leadership team and Murdoch Children’s Research Institute’s Aboriginal Reference Group to ensure ongoing governance and oversight from additional Aboriginal and Torres Strait Islander perspectives and voices, including Victorian Aboriginal community Elders. This review, beyond the initial consultations with stakeholders, does not involve any contact directly with Aboriginal or Torres Strait Islander or other Indigenous program participants within the broader communities or relevant institutions. This research only involves the dissemination of information from previous research conducted with these communities. Throughout the scoping review, ethical principles will be a principal consideration, and relevant guidelines [32,32] will be consulted to ensure the completion of the review to the highest ethical standards in relation to the presentation of information, framing of findings and cultural information.

The proposed review will help to inform future research, policy and practice, specifically with regard to the development of effective therapeutic practices, programs and policies related to healing child sexual abuse.

## Data Availability

The data that support the findings of this study are available from authors J.G. and G.G.

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
