# Peer review of "A Systematic Scoping Review of Indigenous People’s Experience of Healing and Recovery from Child Sexual Abuse"

_ijerph, 2024, doi:10.3390/ijerph21030311_

Round 1
Reviewer 1 Report
Comments and Suggestions for Authors
Manuscript ID: ijerph-281068
A systematic scoping review of Indigenous people’s experience of healing and recovery from Child Sexual Abuse
This manuscript is a protocol paper describing a scoping review designed to address child sexual abuse of Indigenous people to capture traditional practices of healing and recovery as well as barriers and enablers to the disclosure and healing process. The authors are from Australia and shared their efforts to address healing, recovery, and prevention within the Victorian Aboriginal communities. The importance and justification of a scoping review specifically addressing child sexual abuse disclosure and recovery in Indigenous populations is presented in a thoughtful, organized manner, addressing important points of experiences (e.g., colonization) and traditional ways (e.g., healing ceremonies). The authors respectful approach to the topic and desire to integrate the published literature to enhance understanding of traditional healing practices and well-being in Indigenous families is welcomed. The authors present a thorough plan for gathering and summarizing the literature. There are some factors to consider as outlined below.
Below are considerations.
1. The plan for the scoping review is to address a wide range of Indigenous communities, not only in Australia and New Zealand, but also North America, Alaska, Canada, and Europe. The literature review focused primarily on Australia, without integrating what is known about the other Indigenous communities in regards to both experiences of harm and healing practices. As presented it is unclear how the authors understand and will be able to integrate the knowledge, experiences, history, and activities of the range of Indigenous communities.
2. The authors may want to consider widening their search term to capture studies of Tribes in North America. While the term Native American is one that is used, referring to the specific Tribe's name is often preferred. Further, Indian Country and American Indian are common terms in addition to Native American. For example, the work Dr. Dolores Subia Bigfoot, Director of the Indian Country Child Trauma Center, is well respected (e.g., pubmed.ncbi.nlm.nih.gov/20549679/ ). See also the work of Larry EchoHawk on Child Sexual Abuse in Indian Country. There is concern that the current search terms will not completely capture research on child sexual abuse with North American Tribes. I cannot speak to the terminology of Indigenous people’s outside of North America.
3. While most of the references utilize the medical citation process, there are a few articles noted in the texted using the author’s last name and date format, but are not included in the reference list. Note that Black, Frederico and Bamblett (2019) and Milroy, Lawrie & Testro (2018) are cited on page 4, but is not included in the reference list.
4. On page 5, second paragraph, the Joanna Briggs Institute Manual for Evidence Synthesis is cited, but the number appears to be incorrect. It should be 26 rather than 25.
5. On page 7, it states that they will “assess all titles against the inclusion and exclusion criteria.” It would be helpful to clarify what this means, as the titles of the articles would not provide sufficient information to determine if the article fits the inclusion/exclusion criteria for the scoping review.
6. The authors plan to use the Aboriginal and Torres Strait Islander Quality Assessment Tool to examine the extent to which the article is culturally informed and relevant to the lived experiences of Indigenous peoples. Additional information regarding its use and applicability with other Indigenous communities would be important to share.
7. Overall, greater thought about how the literature on the Indigenous communities outside of Australia will be analyzed, understood, integrated, as well as their distinctions respected would helpful. Will there be consultation with elders or other respected individuals in these communities be integrated? How will the lens of the Victorian Aboriginal people’s impact the process?
Author Response
The team would like to thank the reviewer for their feedback. We have also highlighted that we received the following from the editor which we have included in the changes in the manuscript.
|
Editor comments |
|
|
Please also note that for protocol type of article it is necessary also to have section discussion or expected results, so please revise this also during the revision process. For your convenience, I have attached the link for article types where you can check the requirements for protocol type of article. https://www.mdpi.com/about/article_types
|
We thank the editor for highlighting the necessary inclusion of a section for expected results. We have now added in a section in the protocol paper for expected results (see Section 2.14 on lines 460-470), which reads:
|

Reviewer 2 Report
Comments and Suggestions for Authors
This article is well written and the protocol has been well thought through by the authors. It is generally easy to read, makes good use of the literature and presents methodology in a clear way. A few minor points I think will improve the article.
Section 2. Method. I think the reference to 25 should be 26
Research design and throughout: Acronyms make for more difficult reading. consider removing most including MCRI, CSA, QAT, RCT, JBI, PCC, VACCA . Also CSA may tend to inadvertently dehumanise the the nature of child sexual abuse.
2.10 and elsewhere. Please explain RedCAP
2.11 Please explain what you mean by "an Indigenous research methodology stand-point" with appropriate reference.
Reference to the CREATE tool is required. Does that stand for anything?
Author Response
We would like to thank the reviewer for their feedback. Please see the attached for detailed responses. Below we have also detailed feedback to a comment received by the editor, which is also included within the changes.
|
Editor comments |
|
|
Please also note that for protocol type of article it is necessary also to have section discussion or expected results, so please revise this also during the revision process. For your convenience, I have attached the link for article types where you can check the requirements for protocol type of article. https://www.mdpi.com/about/article_types
|
We thank the editor for highlighting the necessary inclusion of a section for expected results. We have now added in a section in the protocol paper for expected results (see Section 2.14 on lines 460-470), which reads:
|

Reviewer 3 Report
Comments and Suggestions for Authors
The authors provide a detailed description of a protocol designed to conduct a systematic scoping review of Indigenous people’s experience of healing and recovery from child sexual abuse (CSA). The primary research questions for the scoping review focus on identifying barriers and enablers to disclosure, access and engagement with services for Indigenous child abuse survivors, and identifying characteristics and components of culturally relevant CSA program, policies, and practices. The authors summarize the current state of knowledge in this area, along with research and practice gaps to be addressed by the review. Given the high prevalence and adverse individual, community and societal consequences of child sexual abuse in Indigenous and non-Indigenous communities globally, findings from the review will help inform future research, therapeutic practices, programs and policies centered in Indigenous healing and recovery. This is an important and understudied area, especially for Indigenous health, and findings from the review will contribute significantly to the scarce literature in this field. Specifically, development of a cultural model for healing child sexual abuse will help move both the research and practice fields forward.
The authors are to be commended on their methodical and participatory approach to protocol development, grounding the scoping review within formative work with partnering agencies and constituents; following systematic approaches (eg, Joanna Briggs Institute (JBI) Manual for Evidence Synthesis) to guide protocol development; Population/Participation, Concept, Context (PCC) criteria to categorize articles and determine eligibility criteria; multiple electronic databases and grey literature to ensure a deep search of relevant peer-reviewed and other relevant literature; PRISMA guidelines to document collection, screening and exclusion processes, and a cultural Risk of Bias and Quality Assessment so that included studies have greater relevance to the lived experience of healing of Indigenous peoples. This inclusion of a cultural quality assessment tool is particularly important for a scoping review specific to Indigenous health. The references included in the protocol manuscript will also be helpful for other researchers interested in conducting a culturally relevant systematic scoping review.
Supplemental materials include detailed inclusion and exclusion criteria which could be modified for similar types of scoping reviews, together with detailed data extraction and quality assessment tools. This level of detail ensures that others could replicate the proposed search with comparable results and provides a model for future systematic scoping reviews to support Indigenous health.
It is rare that I have no comments or specific improvements for a manuscript under review. Instead, I applaud the authors for their systematic and ethical approach to protocol development. Findings from the review will make a significant contribution to knowledge in this area.
Author Response
We would like to thank the reviewer for their feedback. Please see the attached for detailed responses to your comments. We have also detailed below feedback to a comment raised by the editor, which required other changes.
|
Editor comments |
|
|
Please also note that for protocol type of article it is necessary also to have section discussion or expected results, so please revise this also during the revision process. For your convenience, I have attached the link for article types where you can check the requirements for protocol type of article. https://www.mdpi.com/about/article_types
|
We thank the editor for highlighting the necessary inclusion of a section for expected results. We have now added in a section in the protocol paper for expected results (see Section 2.14 on lines 460-470), which reads:
|
